# Effect of Nordic walking on walking ability in patients with peripheral arterial disease: a meta-analysis

Zerong Sun[1], Jing Zhang[1], Yiqun Fang[2], Yongdong Qian [1]*

1 Exercise Human Science Laboratory, College of Physical Education and Health Sciences, Zhejiang Normal University, Jinhua, China, 2 Department of Emergency, Jinhua Guangfu Oncology Hospital, Jinhua, China

* tyxyqyd@zjnu.cn

## Abstract

### Background

Evaluating the effectiveness of Nordic walking in influencing walking ability in patients with peripheral arterial disease.

### Methods

We searched 12 databases, including PubMed, Embase, Cochrane library, Web of Science, EBSCO host, Ovid, Scopus, ClinicalTrial.gov, and several top ranked Chinese databases, including China National Knowledge Infrastructure (CNKI), Wanfang Data, CBMdisc, VIP Database, ChiCTR. The search has no starting time limit and the deadline is April 9, 2024. Randomized controlled trials and pseudo-random controlled trials were included. The two authors independently screened the literature and evaluated the quality of the study using the Cochrane risk of bias tool. Meta analysis was conducted using Review Manager 5.4 and Stata 17.0 software.

### Results

A total of 8 studies involving 508 patients were included. Meta-analysis results showed that compared with supervised exercise therapy (SET), supervised NW was not associated with an increase in maximum walking distance (MWD) and claudication distance (CD) in PAD patients, whether during treadmill tests or 6-minute walk tests (6-MWT), and the results were not statistically significant. In terms of increasing exercise duration, SET was significantly higher than supervised NW and the results were statistically significant (SMD = -0.41, 95% CI: -0.72 to -0.09, Z = 2.54, P = 0.01 < 0.05). Among the 8 studies included, 2 studies had control groups that were not part of the supervised exercise program and were different, therefore no meta-analysis was conducted.

**Data availability statement:** All relevant data are within the paper and its Supporting Information files.

**Funding:** This study was supported by a grant from the Jinhua Science and Technology Research Program Project of Public Welfare Category in 2021 (2021-4-228).

**Competing interests:** The authors have declared that no competing interests exist.

**Abbreviations:** CI, confidence interval; SMD, standard mean difference; NW, Nordic walking; PAD, Peripheral arterial disease.

## Conclusions

In PAD patients, supervised NW is no significant difference in walking ability compared to SET. NW presents a viable option when SET is not available.

## PROSPERO registration

**PROSPERO registration number:** CRD42024535828

## Introduction

Peripheral arterial disease (PAD), a common atherosclerotic condition in cardiovascular disease (CVD), greatly limits patients' future physical abilities. This leads to decreased walking speed, endurance, walking irregularities, and weakened muscle strength [1]. Furthermore, it can potentially increase the risk of CVD-related morbidity and mortality [2]. Intermittent claudication (IC), the classic symptom of PAD, arises from inadequate arterial blood circulation to the lower limbs. It is characterized by cramping or discomfort in the calf, thigh, and/ or buttock during ambulation. Not all PAD patients exhibit typical claudication symptoms, but the primary objective of future PAD treatments remains addressing the decreased walking capacity, diminished quality of life, and compromised balance function observed in PAD patients compared to those without the condition [1,3].

Therapeutic intervention in patients with PAD effectively reduces myocardial infarction, stroke, amputation, and cardiovascular risk [4]. Although specific pharmacological treatments for PAD exist, such as cilostazol (an antiplatelet drug) and Pentoxifylline (peripheral vasodilators), their efficacy in PAD has been limited and inconsistent [5]. Aerobic exercise therapy (ET) and lower-extremity revascularization (LER) are the preferred clinical approaches according to current guidelines. The use of ET has proven to be a beneficial treatment strategy for PAD [6,7].

There exist two main modalities of ET: Supervised Exercise Therapy (SET) and Home-Based Exercise Therapy (HBET). Supervised exercise therapy typically consists of treadmill and walking exercises performed under the direct supervision of a physical therapist [1,5]. HBET consists of a structured aerobic exercise routine primarily done at home without direct supervision [5], providing convenience and reduced hassle [8]. A recent meta-analysis showed that SET significantly outperformed HBET in improving maximal walking distance (MWD) and pain-free walking distance (PFWD) [9]. Despite being effective in alleviating symptoms in IC patients, SET is underutilized, often due to patient fear of exercise-related pain in a hospital setting, as well as practical constraints like transportation costs and time limitations, resulting in low compliance [1,10]. Therefore, alternative training methods such as cycling, lower-extremity resistance training, upper-arm ergometry, and Nordic walking can be valuable options for patients unable to engage in SET [11].

Nordic Walking (NW), which uses a specific pair of cross-country ski pole-like walking poles, in which the shoulder, trunk and abdominal muscles intervene at the same time while walking, has been shown to be very helpful in improving cardiovascular function [12]. NW is an interesting alternative exercise rehabilitation therapy for PAD patients [13–16]. Since various RCTs have been implemented, it is more accurate to report whether separate RCTs generated conflicting evidence, where a meta-analysis can be helpful in settling such controversies.

To date, four meta-analyses of the benefits of NW in patients with PAD have been published. In 2017, Cugusi et al. [17] initially suggested that NW was more favorable in increasing exercise duration and oxygen consumption (Peak VO2) compared to the control group.

However, considering that only two trials were included, their conclusions need to be interpreted with caution. The following year, Golledge et al. [18] investigated the effectiveness of NW in improving walking ability in patients with PAD through a five item controlled trial consisting of three SETs, one HET, and one no exercise routine treatment. The results did not find any significant advantages of NW compared to other treatment methods. Furthermore, Jansen et al. [19] published a high-quality meta-analysis in the Cochrane Library in 2020, which synthesized three NW clinical trials and found no statistically significant differences between the NW group and the control group in maximum walking distance (MWD), pain-free walking distance (PFWD), and the Walking Impairment Questionnaire (WIQ), which is consistent with Golledge et al.'s findings [18]. Recently, a 2023 network meta-analysis [20] included a NW trial that showed no statistically significant difference in improving MWD and 6-minute walking distance (6MWD) compared to the control group.

Although they provide valuable reference information for selecting appropriate exercise plans for PAD patients in clinical practice by comparing different exercise intervention strategies, these four studies still have limitations in the number of included literature. This meta-analysis incorporates more trials based on previous research, and the outcome measures include not only the maximum walking distance (MWD), but also the effectiveness of NW and control groups on claudication distance (CD) and exercise duration in PAD patients. In addition, only the intervention measures and research subjects studied by Golledge et al. [18] were completely focused on NW and PAD patients. Considering that their literature search scope was up to December 2017, which is seven years ago, other scholars may have published new research results during this period, which may provide strong evidence for the comprehensive systematic evaluation of NW. Therefore, the primary objective of this study was to summarize all clinical intervention studies published so far on the impact of NW on the walking ability of PAD patients.

## Methods

This meta-analysis fallowed the Preferred Reporting Items for Systematic Reviews and Meta-Analyses (PRISMA) statement guidelines, and is registered in the PROSPERO (CRD42024535828) On April 14, 2024.

### Selection of research

The inclusion criteria consisted of: (1) Randomized controlled trials (RCT) and pseudo-randomised controlled trials (PRCT) for study design; (2) Peripheral arterial disease and patients with intermittent claudication for study population without restrictions on patients' race, nationality, or disease duration; (3) Nordic walking as the intervention for the trial group with control group programs including supervised exercise therapy, standard home exercise program, and non-exercise routine medical group; (4) Primary outcomes focused on walking ability metrics such as maximum walking distance (MWD), claudication distance (CD), and exercise duration.

The exclusion criteria consisted of: (1) Review and meta-analysis articles; (2) Repetitively published studies; (3) Interventions not related to NW; (4) Intervention groups combining medication with NW; (5) Trials where the control group also had a NW intervention; (6) Incomplete or non-compliant outcome indicators; (7) Abstracts and conference-type studies; (8) Articles discussing non-clinical trials.

### Data sources and searches

A systematic search was conducted using the following databases: China National Knowledge Infrastructure (CNKI), Wanfang Data, CBMdisc, VIP Database, ChiCTR, PubMed, Embase,

Cochrane library, web of science, EBSCO host, Ovid, Scopus and ClinicalTrial.gov. The search has no starting time limit and the deadline is April 9, 2024. The search strategy used a combination of subject headings and free-text terms. All interventional studies on NW in PAD patients were included. Supplementary references were used to enhance access to relevant literature. Additionally, the search strategy for the database is detailed in Supporting information Table 8.

## Data extraction

The two researchers (Z.R.S. and J.Z.) independently screened relevant literature, extracted information, and performed cross-checks. Disagreements were resolved through discussion or consultation with a third researcher. Initial screening involved reading titles and abstracts. After excluding experiments with mismatches, full texts were examined for inclusion eligibility. Articles were extracted for author, year, sample size, study design, study population details, interventions, outcome measures, and indicators. Literature lacking necessary outcome indicators for meta-analysis was excluded.

## Quality assessment

The risk of bias present in the studies included was evaluated independently by two researchers (Z.R.S. and J.Z.), with subsequent cross-verification of their findings. The Cochrane Risk Assessment Tool was employed to appraise the included studies. In the statistical analysis phase, studies were categorized based on their risk of bias: those meeting 5 or more criteria were judged to carry a low risk of bias, those fulfilling 3–4 criteria were assigned a medium risk of bias, and those satisfying fewer than 3 criteria were labeled as high risk.

## Statistical analysis

This meta-analysis compared the effectiveness of clinical interventional studies using NW to treat patients with PAD to improve maximal walking distance (MWD), claudication distance (CD), and exercise duration compared with control patients. First, the PRISMA Flow Diagram was downloaded from the website (http://www.prisma-statement.org/) to create a literature screening flow chart. Schematic diagram of Cochrane bias risk assessment was produced using Review manage5.4. Second, Review manage5.4 and stata 17.0 software were used for statistical processing. Experimental data were continuous variables, and standardized mean difference (STD Mean Difference, SMD) and 95% confidence interval (95% CI) were used as effect scales for combining effect sizes. Effect sizes were categorized as follows: those below 0.2 were deemed small, those falling between 0.2 and 0.8 were considered medium, and those exceeding 0.8 were labeled as large effects. By combining the effect sizes of MWD, CD, and exercise duration using data reported by the experimental and control groups at week 12. If there is no data available at the time of data extraction or the intervention period exceeds week 12, the most recent available time will be used. For example, if the Girold trial [12] only intervenes until week 4, the data at that time point will be used. The intervention period of Collins [21] and Langbein [22] was 24 weeks, and 5 studies [23–27] included in the trial were intervened until 12 weeks. In order to prevent the study time span from being too large, Collins [21] and Langbein's [22] study also used data from the 12th week for meta-analysis. When $P < 0.05$, a statistically significant difference was observed between the experimental and control groups, indicating the statistical validity of the meta-analysis results. Heterogeneity was assessed using Heterogeneity measures (Q-test, α=0.1). Depending on the $I^2$ value obtained from this test, we chose either a fixed-effects model (for $I^2 < 50\%$) or a random-effects model (for $I^2 \geq 50\%$) for the meta-analysis, followed by sensitivity analysis. Finally, the meta-analysis compared MWD, CD, and exercise duration, measured through treadmill tests or 6-minute walk tests, between the NW

and control groups. However, due to the limited availability of studies, we were unable to assess publication bias in the literature. It is worth noting that variations in exercise and testing methods within the NW group may have contributed to heterogeneity in the treadmill test results.

### Inclusion of trial original data conversion

The means and standard deviations of some of the studies in this meta-analysis needed to be transformed from data provided in the literature. For example, the studies by Collins et al. [21], and Langbein et al. [22] provided data on maximum walking time and claudication time on a treadmill at 1.8 mph (equivalent to 2.9 km/h), which were transformed to meters; the study by Spafford et al [24] required the use of the following tool (https://www.math.hkbu.edu.hk/~tongt/papers/median2mean.html) to convert median (range) to Mean±SD; the study by Kropielnicka et al. [25] provided comparative charts of MWD and CD pre- and post-test using Engauge Digitizer software to extract means and standard deviations.

## Results

### Included studies

Initially, 160 publications potentially relevant to this study were collected from 13 databases, as well as one additional publication, for a total of 161 publications. 123 duplicates were eliminated using the literature management software EndnoteX9, and after exclusion by standard screening and full-text reading, 5 RCT trial and 3 non RCT trial articles were selected for qualitative assessment and meta-analysis (Fig 1).

### Trial characteristics

Eight trials were included in this study. The basic characteristics of the included trials are shown in Table 1. The subjects were mostly male, aged 47–80 years. A total of 508 subjects (total sample size ranging from 18-103), with a mixed gender profile, were included. The patient characteristics of the trials are displayed in Table 2. The intervention time range centered on 30–60 minutes. The intervention frequency ranged from 3–5 times per week. There was variation in the intervention period, with the shortest period at 4 weeks and the longest up to 24 weeks. The treadmill test and six-minute walking tests (6-MWT) were primary in measuring MWD, CD, and exercise duration. Six controls were fully supervised therapies, while the other two were the conventional treatment and standard home exercise therapy (HET). Of the trials in which the control group was SET, the control groups of Collins et al. [21], Collins et al. (exercise duration) [23] and Girold et al. [12] were all conventional walking exercise, Bulinska et al. [27], Kropielnicka et al. [25] and Dziubek et al. [26] were all treadmill exercise. For the remaining two trials, one intervention group was three supervised Nordic walking sessions per week combined with one unsupervised but guided exercise session, and the control group received the best medical treatment alone [22]. One was an unsupervised Nordic walking program based on home exercise, the control group was an unsupervised standard home exercise therapy with written instructions, and both groups were supported by pedometers, diaries, and weekly phone calls from a physical therapist [24]. In this study, the main focus was on changes in MWD, CD and exercise duration, so only the corresponding indicators were extracted.

### Quality assessment

The results of the quality assessment are shown in Fig 2 and Fig 3. In the RCT, 4 articles reached low risk of bias and had high quality, while the remaining 1 articles had moderate bias; In Fig 2, if it meets the standard, it is marked as "+", and if it does not meet the standard,

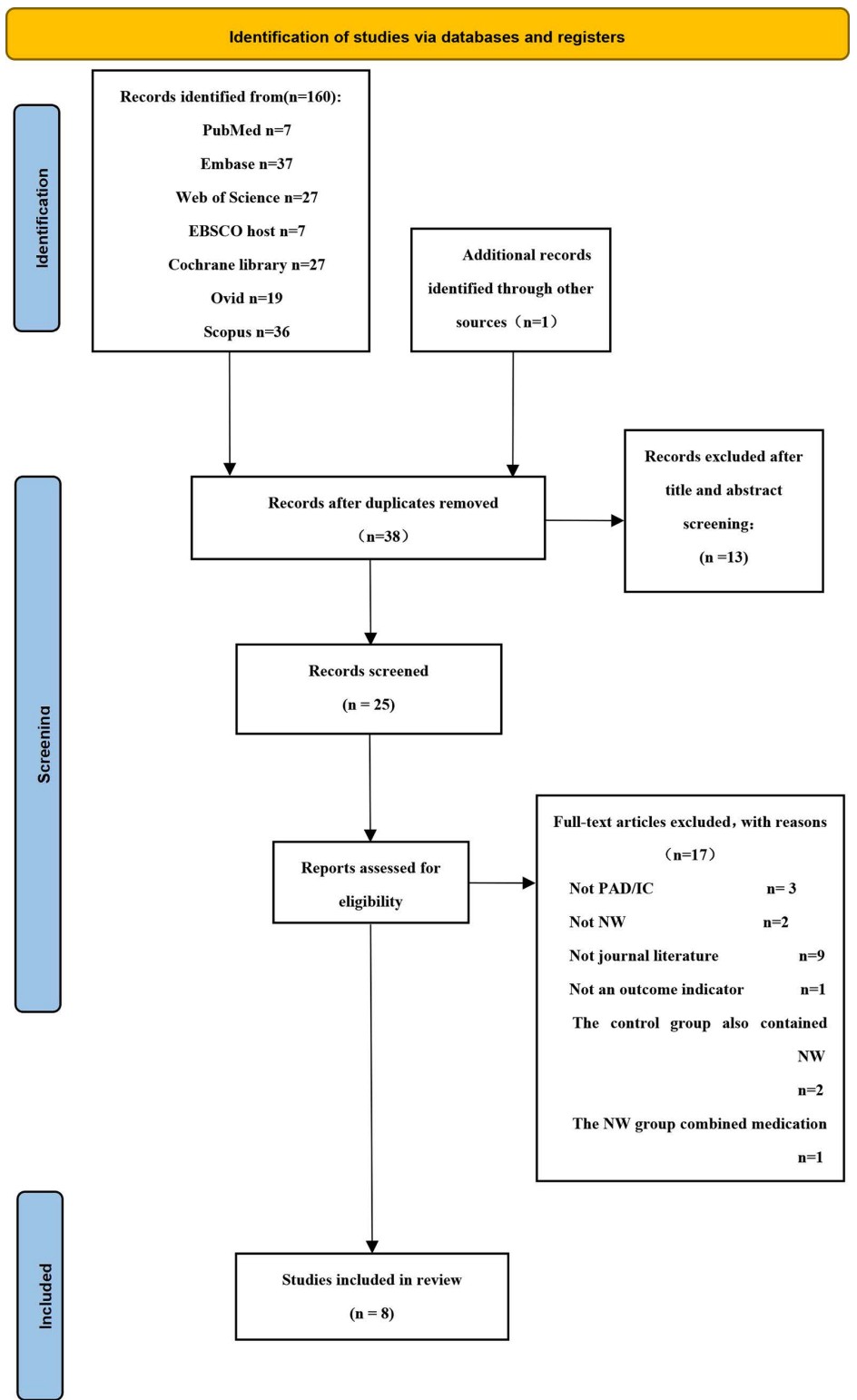

**Fig 1. A PRISMA flowchart illustrating the process of study identification.** PAD peripheral artery disease, IC intermittent claudication, NW Nordic walking.

**Table 1. Characteristics of the trials that were included. (N = 8).**

| Study | Type | N | Baseline N | Completed study N | Group | Intervention period (weeks) | Frequency (per week) | Duration (minutes) | Outcome assessment | Outcome indicator |
|---|---|---|---|---|---|---|---|---|---|---|
| Collins 2012 [21] | RCT | 103 | 51 | 43 | Supervised NW | 24 | 3 | 30-60 | Treadmill walking test (Symptom-limited [c], Constant work rate sub maximal treadmill test protocol [b]) | MWD, CD, exercise duration |
| | | | 52 | 46 | Supervised SET | 24 | 3 | 30-60 | | |
| Langbein 2002 [22] | RCT | 52 | 27 | 27 | Supervised NW | 24 | 4 | 30-45 | Treadmill walking test (Symptom-limited [b], Constant work rate sub maximal treadmill test protocol [b]) | MWD, exercise duration |
| | | | 25 | 25 | Non exercise | 24 | NA | NA | | |
| Collins (duration) 2012 [23] | RCT | 85 | 45 | 35 | Supervised NW | 12 | 3 | 30-55 | Treadmill walking test (Symptom-Limited Treadmill Test Protocol [c]) | Exercise duration |
| | | | 40 | 36 | Supervised SET | 12 | 3 | 30-55 | | |
| Spafford 2014 [24] | RCT | 52 | 28 | 19 | Unsupervised NW | 12 | 3 | 30 | 6-MWT | MWD, CD |
| | | | 24 | 19 | Unsupervised HET | 12 | 3 | 30 | | |
| Girold 2017 [12] | RCT | 18 | 9 | 9 | Supervised NW | 4 | 5 | 45 | 6-MWT, Treadmill walking test (modified Bruce protocol) | MWD |
| | | | 9 | 9 | Supervised SET | 4 | 5 | 45 | Treadmill walking test (Gardner–Skinner protocol [a]), 6-MWT | MWD, CD |
| Bulinska 2016 [27] | PRCT | 70 | 35 | 21 | Supervised NW | 12 | 3 | 30-50 | Treadmill walking (Gardner–Skinner protocol [a]), 6-MWT | MWD, CD |
| | | | 35 | 31 | Supervised SET | 12 | 3 | 30-50 | 6-MWT | MWD, CD |
| Kropielnicka 2018 [25] | PRCT | 64 | 32 | 21 | Supervised NW | 12 | 3 | 45 | 6-MWT, Treadmill walking test (modified Bruce protocol) | MWD |
| | | | 32 | 31 | Supervised SET | 12 | 3 | 45 | Treadmill walking test (Gardner–Skinner protocol [a]), 6-MWT | MWD, CD |
| Dziubek 2020 [26] | PRCT | 64 | 32 | 21 | Supervised NW | 12 | 3 | 45 | Treadmill walking (Gardner–Skinner protocol [a]), 6-MWT | MWD, CD |
| | | | 32 | 31 | Supervised SET | 12 | 3 | 45 | 6-MWT | MWD, CD |

MWD Maximum walking distance, CD claudication distance, RCT randomized controlled trial, PRCT pseudo-randomised controlled trial, 6-MWT six minute walking tests, NW Nordic Walking, SET Supervised Exercise Therapy, HET standard home exercise therapy, NA Not applicable.

Two experiments in this study were first authored by Collins and both were published in 2012. The first experiment provided data on MWD and CD on a treadmill test [21], while the second experiment provided the longest exercise duration for patients to walk on the treadmill, therefore represented by Collins et al. (exercise duration) [23].

[a]The treadmill test maintained a constant velocity of 3.2 km/h, while gradually elevating the incline by 2% every 2 minutes.

[b]The treadmill test maintained a constant velocity of 2.9 km/h alongside a fixed incline of 12%.

[c]The treadmill test maintained the percent grade increasing every 30 seconds and the velocity increasing every 3 minutes after the first 6 minutes(amount not stated precisely).

it is marked as "-". All included trials obtained approval from the local ethics committee in accordance with legal regulations. Participants provided written informed consent forms before enrollment, record their height and weight, and teach how to use Nordic walking poles. Therefore, blinding cannot be used for both participants and assessors, and only one trial (Girold et al. [12]) reported blinding by the outcome assessors. Many trials allowed 20% of subjects to drop out, with Bulinska et al. [27], Kropielnicka et al. [25], Spafford et al. [24] and Dziubek et al. [26] having dropout rates of 40%, 34%, 32%, and 34% in the NW group, respectively. The large difference in size between the two groups is a major limitation.

## Reported outcomes

The main outcome measures reported in this study encompassed MWD (maximum walking distance), CD (claudication distance), and exercise duration. S1 Data present the sample size, mean, and standard deviation data corresponding to MWD, CD, and exercise duration

**Table 2. Characteristics of patients recruited to the included trials. (N = 508).**

| Study | Sex Group (Males/Females) | Age Group (Mean±SD) | ABI Group (Mean±SD) | Inclusion of the patient's leg level | BMI (Mean±SD) | Current Smoking N (%) | Diabetes N (%) | Airway disease N (%) | CHD N (%) | Hypertension N (%) | Dyslipidaemia N (%) |
|---|---|---|---|---|---|---|---|---|---|---|---|
| Collins 2012 [21] | T 47/4 | T 71.4 ± 9.1 | T 0.62 ± 0.20 | Subjects with ABI ≤ 0.90 or documented vascular calcification in the most severely affected leg. | 29.0 ± 4.4 | 35(68.6) | NR | NR | NR | NR | NR |
| | C 49/3 | C 68.0 ± 8.5 | C 0.65 ± 0.36 | | 28.6 ± 5.5 | 35(67.3) | | | | | |
| Langbein 2002 [22] | Total 51 males, 1 female | T 65.5 ± 6.8 | T 0.64 ± 0.25 or | IC, ABI <0.95 at rest or <0.85 after exercise. | 28.6 ± 4.9 | 10(37) | NR | NR | NR | NR | NR |
| | | C 68.7 ± 8.8 | C 0.69 ± 0.14 | | 28.4 ± 5.4 | 9(36) | | | | | |
| Collins (duration)2012 [23] | T 41/4 | T 71.7 ± 9.2 | T 0.62 ± 0.20 | ABI ≤0.90 | 29.2 ± 4.4 | 16(36) | 21(47) | NR | NR | 35(78) | 34(76) |
| | C 38/2 | C 66.8 ± 8.5 | C 0.63 ± 0.17 | | 28.9 ± 5.5 | 13(33) | 20(52) | | | 36(92) | 29(73) |
| Spafford 2014 [24] | T 19/9 | T 65.0 ± 10.6 | T 0.61 ± 0.16 | Stable IC for at least 6 months with [a] resting ABI <0.9. | 28.0 ± 5.3 | 4(14.3) | 7(25.0) | 4(14.3) | 6(21.4) | 15(53.6) | 26(92.9) |
| | C 16/8 | C 65.0 ± 9.8 | C 0.61 ± 0.20 | | 29.0 ± 4.9 | 5(20.8) | 9(37.5) | 4(16.7) | 5(20.8) | 16(66.7) | 21(87.5) |
| Girold 2017 [12] | NR | T 58.2 ± 10.4 | NR | NR | 28.2 ± 4.8 | NR | NR | NR | NR | NR | NR |
| | | C 60.9 ± 11.3 | | | 27.0 ± 4.7 | | | | | | |
| Bulinska 2016 [27] | T 12/9 | T 67.0 ± 9.3 | T 0.71 ± 0.22 | ABI < 0.9, claudication pain while walking | 26.8 ± 4.3 | NR | 8(38.1) | NR | 10(47.6) | 12(57.1) | 9(42.9) |
| | C 25/6 | C 67.0 ± 7.4 | C 0.68 ± 0.19 | | 27.9 ± 3.7 | | 12(38.7) | | 10(32.3) | 27(87.1) | 19(61.3) |
| Kropielnicka 2018 [25] | NR | T 67.00 ± 9.32 | T R 0.76 ± 0.17 L 0.71 ± 0.22 | PAD with lower limb ischemia and IC over distances of 30-400 meters, ABI <0.9 | 26.77 ± 4.34 | NR | NR | NR | NR | NR | NR |
| | | C 67.00 ± 7.43 | C R 0.68 ± 0.19 L 0.68 ± 0.16 | | 27.85 ± 3.72 | | | | | | |
| Dziubek 2020 [26] | NR | T 67.00 ± 9.32 | T R 0.76 ± 0.17 L 0.71 ± 0.22 | CD stabilized for at least 3 months, ABI <0.9 | NR | 9(42.9) | 8(38.1) | NR | NR | 14(66.7) | NR |
| | | C 67.00 ± 7.43 | C R 0.68 ± 0.19 L 0.68 ± 0.16 | | | 9(29) | 12(38.7) | | | 26(83.9) | |

NR not reported, T trials groups, C control groups, R Right leg, L left leg, IC intermittent claudication, CHD coronary heart disease, CD claudication distance, NW Nordic Walking, SET Supervised Exercise Therapy, HET standard home exercise therapy

The data are presented as either percentages (%) or means (± standard deviation).

(Sex, age, ABI, BMI) Mean±SD in parentheses; (Smoking, diabetes, Airway disease, Coronary heart disease, Hypertension, Dyslipidemia) percent of total in parentheses.

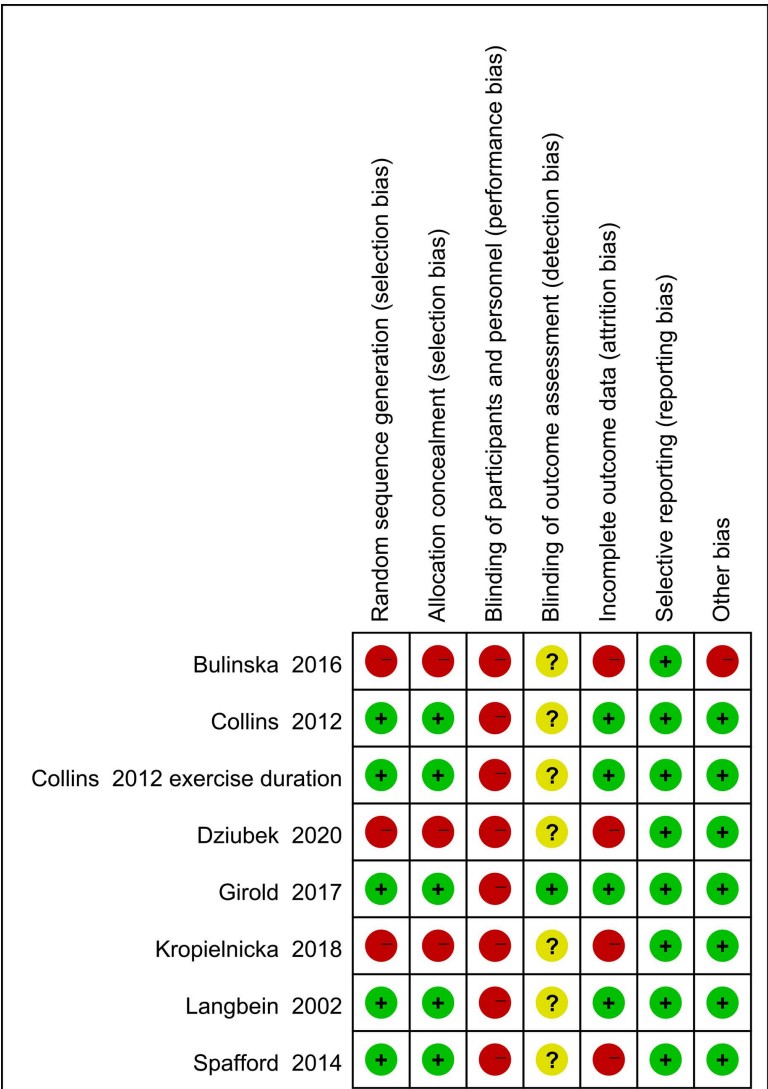

**Fig 2. Schematic representation of the methodological quality evaluation of the literature in this study.**

for each individual trial. Among the 8 intervention studies included, 6 studies had a supervised exercise program as the control group, while 2 studies had completely different control groups, including non exercise conventional treatment and unsupervised home exercise therapy, and their data could not be merged for meta-analysis. Therefore, the final 6 studies were included.

## Effect of supervised NW versus control/supervised exercise therapy

**Maximal walking distance (MWD).** The control groups across the five trials encompassed in this study all underwent supervised exercise therapy (SET), primarily assessing MWD and CD through treadmill tests and 6-minute walk tests (6-MWT). Among the four trials that used treadmill tests, significant heterogeneity was observed, as indicated by an $I^2 = 60\%$ (>50%), $P < 0.1$ in the Q test. This heterogeneity suggests strong variation among the studies included in our analysis. Consequently, a random effects model was chosen for the meta-analysis. The

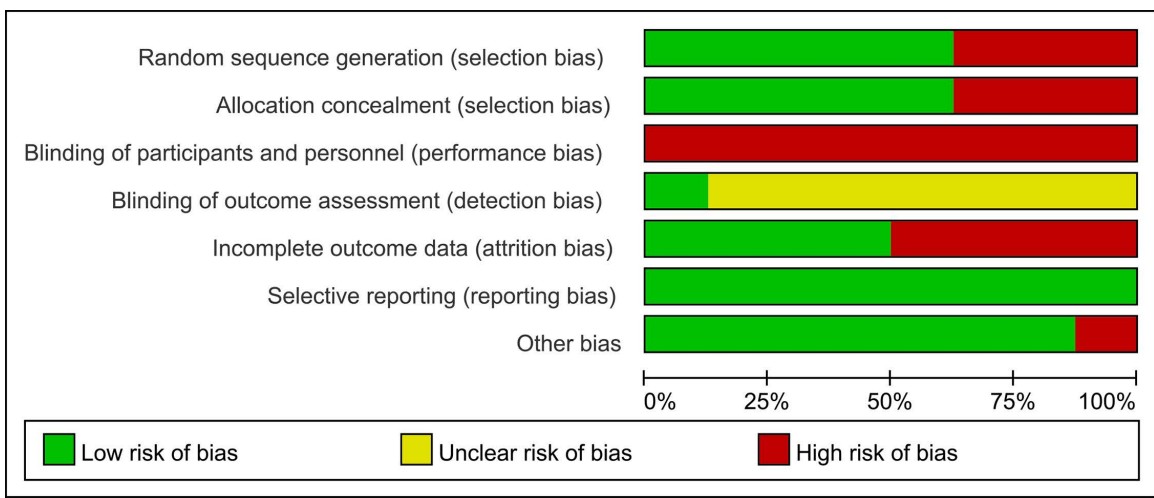

**Fig 3. Percentage of each of the methodological quality evaluation of the literature in this study.**

findings revealed that supervised Nordic walking did not demonstrate a statistically significant improvement in increasing the MWD for patients with PAD compared to SET (SMD = -0.06, 95% Cl: -0.52 to 0.41, Z = 0.24, P = 0.81 > 0.05) (Fig 4).

After conducting the heterogeneity test for the four trials utilizing the 6-Minute Walk Test (6-MWT), it was observed that there was no heterogeneity, with $I^2 = 0\%$ (<50%), P > 0.1 for the Q-test. This finding indicates homogeneity among the selected trials for this study. As a result, the fixed effect model was chosen for meta-analysis. However, although the supervised NW group demonstrated a marginal improvement over SET in terms of increasing the MWD, the difference did not reach statistical significance (SMD = 0.24, 95% Cl: -0.06 to 0.55, Z = 1.57, P = 0.12 > 0.05) (Fig 5).

**Claudication distance (CD).** The four trials encompassed in this study primarily relied on the treadmill test and the 6-MWT to assess CD outcomes. For the three studies using the treadmill test, heterogeneity testing revealed an $I^2 = 0\%$, P > 0.1 for the Q test. This indicates that there is no heterogeneity between the selected experiments, confirming homogeneity. Based on these findings, we chose a fixed effects model for meta-analysis. The results showed that the difference in CD between the SET group and the supervised NW group did not reach statistical significance (SMD = 0.17, 95% Cl: -0.12 to 0.45, Z = 1.14, P = 0.25 > 0.05) (Fig 6).

For the two studies using the 6-MWT, heterogeneity was tested. The results indicated homogeneity between the selected trials, with an $I^2 = 0\%$ (< 50%), P > 0.1 in the Q-test. Based on these findings, a fixed-effects model was adopted for the meta-analysis. The results showed that the difference in CD between the SET group and the supervised NW group did not reach statistical significance (SMD = -0.09, 95% Cl: -0.49 to 0.30, Z = 0.47, P = 0.64 > 0.05) (Fig 7).

**Exercise duration.** The two trials encompassed in this study used a treadmill test to assess exercise duration. Following a heterogeneity assessment, with an $I^2 = 0\%$ (< 50%) and P > 0.1 in the Q-test, it was evident that there was no heterogeneity among the selected trials for this study. Consequently, the fixed effect model was chosen for meta-analysis. The findings indicated that SET notably surpassed supervised NW in prolonging the exercise duration of PAD patients, and the meta-analysis results were statistically significant (SMD = -0.41, 95%Cl: -0.72 to -0.09, Z = 2.54, P = 0.01 < 0.05) (Fig 8).

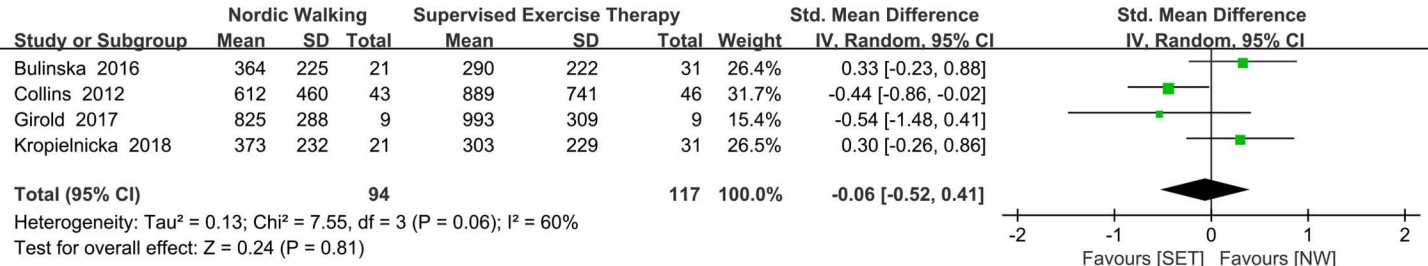

**Fig 4. Results of a meta-analysis comparing maximum walking distance in treadmill testing between supervised NW and supervised exercise therapy.** In the study by Bulinska et al. [23], at 12 weeks of intervention there were 43 in the NW group and 46 in the control group; at the end of the intervention (24 weeks) there were 34 in the NW group and 43 in the control group.

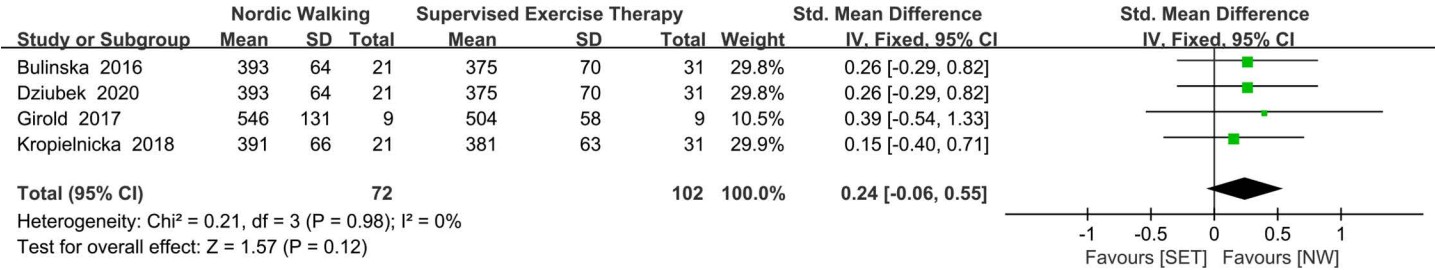

**Fig 5. Results of a meta-analysis comparing maximum walking distance in 6-MWT between supervised NW and supervised exercise therapy.**

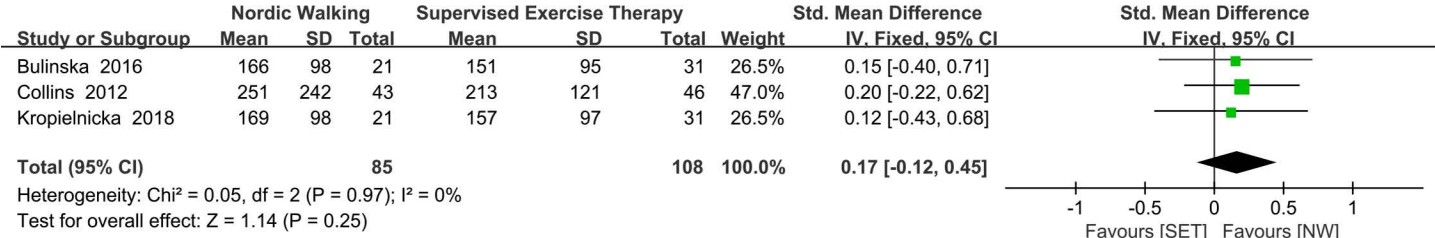

**Fig 6. Results of a meta-analysis comparing claudication distance in treadmill testing between supervised NW and supervised exercise therapy.** In the study by Collins et al. [23], at 12 weeks of intervention there were 43 in the NW group and 46 in the control group; at the end of the intervention (24 weeks) there were 34 in the NW group and 43 in the control group.

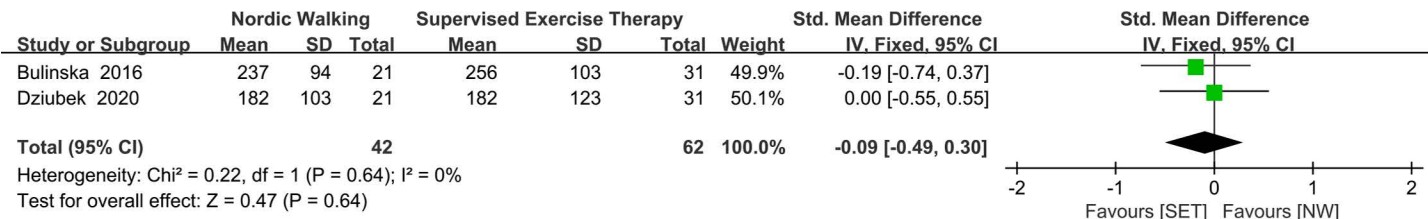

**Fig 7. Results of a meta-analysis comparing claudication distance in 6-MWT between supervised NW and supervised exercise therapy.**

| Study or Subgroup | Nordic Walking Mean | SD | Total | Supervised Exercise Therapy Mean | SD | Total | Weight | Std. Mean Difference IV, Fixed, 95% CI | Std. Mean Difference IV, Fixed, 95% CI |
|---|---|---|---|---|---|---|---|---|---|
| Collins 2012 | 12.67 | 9.52 | 43 | 18.39 | 15.34 | 46 | 55.4% | -0.44 [-0.86, -0.02] | |
| Collins 2012 exercise duration | 10.83 | 3.77 | 35 | 12.31 | 4.27 | 36 | 44.6% | -0.36 [-0.83, 0.11] | |
| **Total (95% CI)** | | | 78 | | | 82 | 100.0% | -0.41 [-0.72, -0.09] | |

Heterogeneity: Chi² = 0.06, df = 1 (P = 0.81); I² = 0%
Test for overall effect: Z = 2.54 (P = 0.01)

**Fig 8. Results of a meta-analysis comparing exercise duration in treadmill testing between supervised NW and supervised exercise therapy.** In the study by Collins et al. [23], at 12 weeks of intervention there were 43 in the NW group and 46 in the control group; at the end of the intervention (24 weeks) there were 34 in the NW group and 43 in the control group.

## Sensitivity analysis

The sensitivity analyses for MWD and CD were based on all eight studies, and as can be seen in (Figs 9 and 10), there was little difference in heterogeneity between studies, and the exclusion of a particular article did not have much effect on the effect sizes, and the findings of the review were relatively stable.

## Discussion

This study aims to integrate intervention studies on the impact of NW on patients with PAD and IC symptoms. The four previously published meta-analyses [17–20] have certain limitations in their research methods due to the limited number of included literature. Firstly, Golledge et al.'s meta-analysis [18] was unable to subdivide the control group, but instead mixed them together for analysis. When evaluating walking ability, the analysis was based on the baseline number of participants and did not exclude participants who withdrew from the study midway. The outcome measure only involved the maximum walking distance, and did not directly explore the claudication distance and exercise duration. Its results were similar to those of Jansen et al. [19] and Tremblay et al. [20], both indicating that NW had no significant advantage compared to the control group. It is worth noting that Jansen et al.'s study [19] has high reference value due to its publication in the Cochrane Library. The systematic reviews published by the Cochrane Library ensure the high quality and authority of research due to their adherence to strict standardization processes, regular updates, and continuous improvement feedback mechanisms. Secondly, Cugusi et al. [17] confirmed that traditional walking (SET) has greater improvements in exercise time (ExD) and peak oxygen uptake (VO2) compared to NW; Compared to other treatment methods, NW is more effective. But other treatment methods only include two trials, so they cannot provide reliable conclusions. Finally, apart from Golledge et al.'s study [18], the other three meta-analyses have a broader inclusion of research subjects and intervention measures. Cugusi et al.'s study [17] covered all randomized controlled trials prior to November 2016, including not only PAD patients but also patients with other types of cardiovascular disease. Jansen [19] and Tremblay's study [20], although targeting the population of PAD patients, focused on evaluating the efficacy of different exercise methods in improving patients' walking ability. The quality and potential bias of the trials included in this study are considered acceptable. Therefore, current information can assist therapists and researchers interested in practicing Nordic walking as a rehabilitation exercise method.

In this review, we examined how Nordic walking affects the walking ability of patients with PAD from an evidence-based perspective. The meta-analysis findings indicate that compared with SET, supervised NW was not associated with an increase in MWD and CD in PAD patients, whether during treadmill tests or 6-MWT. In terms of increasing exercise

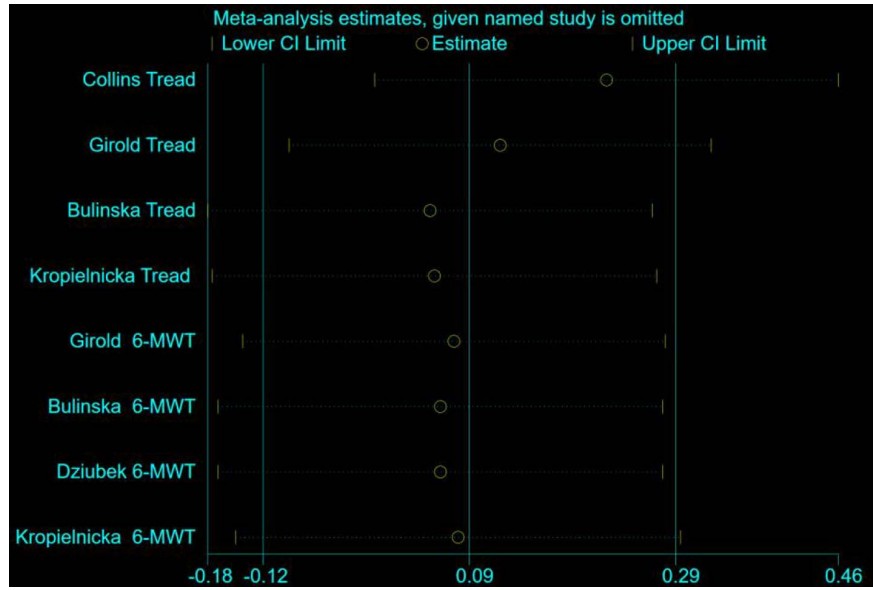

**Fig 9. Sensitivity analysis of NW versus controls for MWD among people with PAD.**

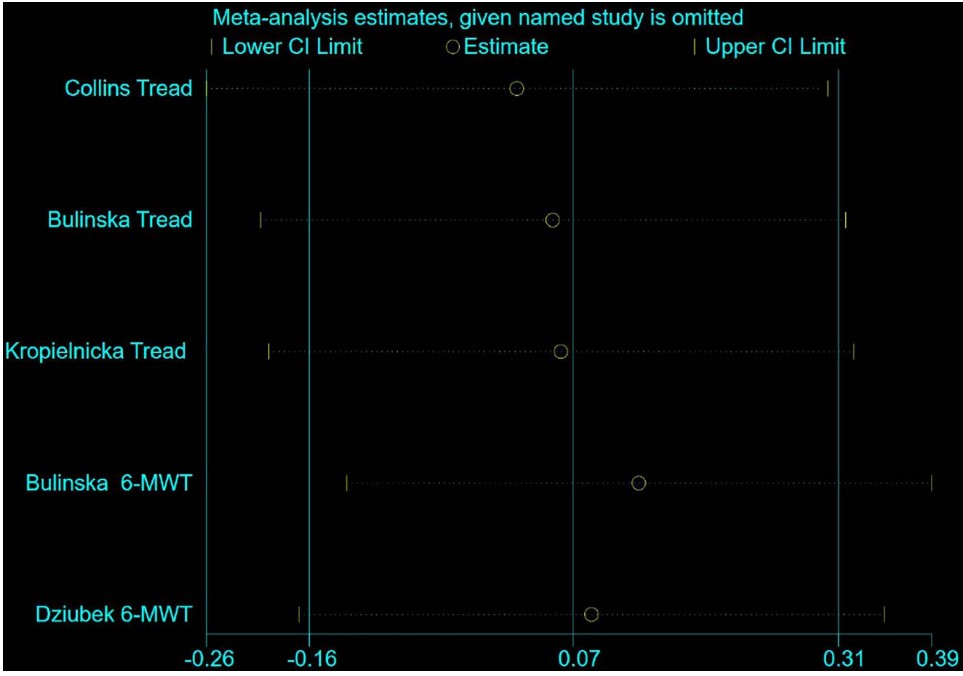

**Fig 10. Sensitivity analysis of NW versus controls for CD among people with PAD.**

duration, SET was significantly higher than supervised NW, consistent with the results of the Cugusi et al. Study [17]. Among the 8 included studies [12,21–27], 2 studies [22,24] had a control group that did not belong to the supervised exercise program (SET), therefore no meta-analysis was conducted. The results of both trials were more supportive of NW. Langbein et al. [22] used supervised NW in the intervention group, while the control group

received routine treatment without exercise. Spafford et al. [24] used unsupervised NW in the intervention group, while the control group received unsupervised home exercise therapy. Regardless of whether the intervention group was conducted in a directly supervised environment, they were more effective than the control group in increasing MWD and improving walking function in PAD patients. It is worth noting that two studies [22,24] from intervention to week 12 showed that in terms of increasing patient MWD, supervised NW was significantly better than unsupervised NW (1001 ± 913 m vs 652 ± 373 m). This may mean that the extent to which NW intervention improves patient MWD compared to other treatment methods (except SET) is influenced by the monitoring itself, rather than the exercise style.

Supervised exercise program is a widely recognized first-line treatment for PAD walking ability, which has been shown to increase the onset time of claudication (COT) and peak walking time (PWT) [28]. In 2007, Wind et al. conducted a systematic review comparing SET with unsupervised exercise therapy (exercise guidance) and found that SET can increase the pain-free walking distance (PFWD) and MWD of patients with intermittent claudication [29]. The results were more supportive of SET (WMD = 143.81, 95% Cl: 5.81 to 281.81, Z = 2.04, P = 0.04); (WMD = 250.40, 95% Cl: 192.35 to 308.45, Z = 8.45, P < 0.00001), But whether it is more effective than exercise guidance requires further research. Another meta-analysis [30] published by Fokkenrood et al. in Cochrane library in 2013 also showed that after three months of intervention, participants who participated in supervised walking exercise walked 180 meters more than those who participated in unsupervised walking exercise, which improved their walking ability (MWD and PFWD) to a greater extent (SMD = 0.69, 95% Cl: 0.51 to 0.86, Z = 7.80, P < 0.00001); (SMD = 0.70, 95% Cl: 0.52 to 0.89, Z = 7.37, P < 0.00001). In the same year, Parmenter et al. [31] compared supervised whole body high-intensity PRT (H-PRT) with routine treatment of unsupervised walking for 6 months and found that H-PRT significantly improved the 6-minute walking distance (6MWD) of elderly people with intermittent claudication symptoms, while unsupervised walking exercise did not (n = 8, MD = 62.6 ± 58.0 m, P = 0.02, n = 7, MD = -9.9 ± 52.9 m). The above studies all show that the supervised exercise group has greater statistical benefits compared to the unsupervised exercise group. However, SET cannot be widely used in the treatment of PAD [4,32]. Firstly, the cost of SET is high, and medical insurance does not reimburse such exercise expenses, which is an economic burden for PAD patients. Secondly, patients need to go to the hospital for exercise three times a week, especially for PAD patients with limited mobility, which is a significant expense. In addition, most doctors and patients lack understanding of SET. Even if some studies provide free treatment to patients, many people still choose to refuse [33]. It is difficult for elderly people to maintain their walking speed while walking on a treadmill, and they often grip the treadmill tightly for fear of falling, so they rarely exercise their upper limbs [21]. In a 2023 trial of the feasibility of using a supervised exercise program and high-intensity interval training program (HIIT) for the treatment of IC patients, it was found that compliance and completion rates of the supervised exercise program were low, with 165 eligible for clinical trials (59%), but only 40 patients (24%) agreed to participate in the trial [34]. Unsupervised exercise programs are attractive to PAD patients, partly because many patients are unable to undergo SET. Home Exercise Therapy (HET) is a cost-effective and convenient alternative to SET [33]. Its intervention measures include not only "walking home", but also regular phone support and encouragement from coaches, and requiring patients to record their exercise and monitor their physical activity. For many PAD patients, HET with convenient exercise methods and low prices are easier for them to accept [33], and have significant advantages compared to non exercise conventional treatments. In a randomized clinical trial [35] conducted by McDermott et al. in 2021, participants with PAD

were randomly divided into three groups: low-intensity walking exercise (n = 116), high-intensity walking exercise (n = 124), and non exercise control group (n = 65). Both exercise groups were required to conduct walking exercises in an unsupervised environment. The results showed that the effect of low-intensity family exercise was significantly lower than that of high-intensity family exercise, and there was no significant difference in improving 6-minute walking distance (6MWD) compared to non exercise control. The 6MWD of the high-intensity exercise group increased from 338.1 m to 371.2 m (within-group mean change = 34.5 m, 95% Cl: 20.1 to 48.9 m, P < 0.001); non exercise control group increased from 328.1 m at baseline to 317.5 m at the 12-month follow-up (within-group mean change = -15.1 m, 95% Cl: -35.8 to 5.7 m, P = 0.10). In summary, both supervised and unsupervised exercises can effectively improve the walking ability of PAD.

In this study, we found that there was no statistically difference in the improvement of walking ability between the supervised NW group and the SET group in PAD patients, indicating that supervised NW alone use did not significantly surpass SET in improving patients' walking ability. This discovery may suggest that the two therapies have similar effects on walking rehabilitation for PAD patients. Therefore, we can reasonably infer that supervised NW is also an effective alternative treatment option for PAD patients. Compared with SET and HET, it is more innovative. In addition to being able to perform under supervised and unsupervised conditions to alleviate patients' tension, it also has the following advantages. Firstly, NW has the advantages of being low-cost, safe and easy to learn [13]. Secondly, it strengthens one's core and upper limb muscles, improves oxygen consumption (VO2) and walking speed, and burns excess fat during exercise and is less prone to fatigue than SET and resistance training [14]. Lastly, there is evidence to suggest that NW, compared with normal walking, when walking on uphill slopes and flat surfaces, can reduce the burden on the spine and legs, as people with PAD symptoms also suffer from spinal, hip and knee arthritis [15], making it an appropriate form of exercise for the elderly population [16]. There are numerous clinical trials that have confirmed the effectiveness of NW in the PAD population. The study by Oakley et al. [36] included 21 male participants and compared NW with normal walking. The results showed that after intervention, CD increased from 133 ± 127 m to 258 ± 278 m and MWD increased from 317 ± 267 m to 377 ± 260 m in the NW group. However, as the control group data was not clearly provided in the study, it could not be included in this meta-analysis. Another clinical randomized controlled trial [37] also confirmed that NW significantly improved exercise tolerance in constant work rate and progressive symptom-limited treadmill testing. The results showed that in the progressive symptom limited treadmill test, the exercise duration of participants in the NW+vitamin group increased from baseline 10.2 ± 4.47 min to 15.9 ± 3.55 min after 6 months of intervention; The NW+placebo group increased from 10.65 ± 3.88 min to 15.57 ± 5.42 min; The vitamin group increased from 11.9 ± 5.35 min to 10.13 ± 5.32 min; The placebo group increased from 10.18 ± 3.9 min to 10.57 ± 4.02 min; In the constant working rate treadmill test, the exercise time of subjects in the NW+vitamin group increased from baseline 7.77 ± 6.00 min to 31.43 ± 18.2 min after 6 months of intervention; The NW+placebo group increased from 13.40 ± 10.67 min to 33.70 ± 23.15 min; The vitamin group decreased from 11.75 ± 15.85 min to 11.06 ± 9.13 min; The placebo group increased from 10.20 ± 6.30 min to 10.38 ± 8.85 min. Due to the combination of vitamins and placebo in the NW group of this experiment, it was excluded from this study, and since supplementing with vitamin E has almost no additional benefits on exercise ability, the results of this study are worth referring to. In addition, from the compliance of the subjects included in this study, it was found that the overall compliance was good, but the NW group had the highest number of people who withdrew midway. The main reason may be that the subjects often face unfavorable weather conditions when using Nordic

walking poles for exercise, and are difficult to persist in outdoor activities due to deteriorating health conditions. They are also influenced by factors such as their own psychology and long round-trip distances.

Limitations of this study: (1) The quantity of incorporated literature is limited, and certain studies possess a modest sample size, wherein a mere 18 participants were enrolled in a single experiment [12]. This, combined with the fact that some participants withdrew from the trial due to health problems or personal reasons, can result in a greater risk of bias. (2) The intervention duration of the studies encompassed varied from a minimum of 4 weeks to a maximum of 24 weeks, and the present study synthesized effect sizes with the data reported in the 12th week, and some of the studies lacked the week 12 data and can only be synthesized with data from the most recent cycle, with a higher risk of implementation bias. (3) The sample size consisted of older males, which may not be applicable to younger populations with a higher proportion of females. (4) The quality assessment of this study has a certain degree of subjectivity, and the Mean and standard deviation of some studies need to be transformed, which may result in errors and potentially affect the research results. (5) There was a obvious heterogeneity in the study protocols of the included trials. For example, the heterogeneity of the MWD in the treadmill test compared with SET was moderately heterogeneous (60%), and the heterogeneity between the other groups was 0%, which may be due to the different methods of treadmill testing, with some of the trials using the constant work rate treadmill test, and the rest using the graded treadmill test. (6) The control group included in the study, apart from the supervised exercise program, only consisted of two trials with different treatment regimens, making it impossible to conduct a meta-analysis. We look forward to more relevant studies in the future to expand the gap in this area of research. Finally, there were some methodological issues with the inclusion of trials, such as three trials were non RCT studies, four trials had more than a 20% subject dropout rate, and only one used evaluator blinding and did not report economic benefits.

## Conclusions

Nordic walking can improve the walking ability of PAD patients. However, compared to standard exercise therapy (SET), supervised Nordic walking does not have significant advantages; SET was more effective in enhancing exercise duration. Nordic walking presents a viable option when SET is not available. Further high-quality research is needed due to the limited number of studies to validate these findings.

## Supporting information

**S1 Checklist. PRISMA_2020_checklist.**
(DOCX)

**S1 Data. Supplementary data of studies included.**
(DOCX)

**S1 File. Excluded studies and reason(N = 153).**
(DOCX)

**S2 File. Quality assessment of randomized controlled trials and pseudo-random controlled trials using Cochrane risk of bias tool.**
(DOCX)

**S3 File. Detailed data of studies included.**
(XLSX)

**S4 File. PICOS principle.**
(DOCX)

**S5 File. The search and screening strategy.**
(DOCX)

## Author contributions

**Conceptualization:** Zerong Sun.

**Data curation:** Zerong Sun, Jing Zhang.

**Formal analysis:** Zerong Sun.

**Funding acquisition:** Yiqun Fang, Yongdong Qian.

**Investigation:** Zerong Sun, Jing Zhang.

**Methodology:** Zerong Sun.

**Software:** Zerong Sun.

**Supervision:** Yiqun Fang, Yongdong Qian.

**Validation:** Yiqun Fang, Yongdong Qian.

**Writing – original draft:** Zerong Sun.

**Writing – review & editing:** Yiqun Fang, Yongdong Qian.

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
