## [Decision Letter · Decision Letter 0]

16 Aug 2024

PONE-D-24-29043Effect of Nordic Walking on Walking Ability in Patients with Peripheral Arterial Disease: A Meta-AnalysisPLOS ONE

Dear Dr. Qian,

Thank you for submitting your manuscript to PLOS ONE. After careful consideration, we feel that it has merit but does not fully meet PLOS ONE’s publication criteria as it currently stands. Therefore, we invite you to submit a revised version of the manuscript that addresses the points raised during the review process.

We look forward to receiving your revised manuscript.

Kind regards,

Shukri AlSaif

Academic Editor

PLOS ONE

Journal Requirements:

2. PLOS requires an ORCID iD for the corresponding author in Editorial Manager on papers submitted after December 6th, 2016. Please ensure that you have an ORCID iD and that it is validated in Editorial Manager. To do this, go to ‘Update my Information’ (in the upper left-hand corner of the main menu), and click on the Fetch/Validate link next to the ORCID field. This will take you to the ORCID site and allow you to create a new iD or authenticate a pre-existing iD in Editorial Manager. Please see the following video for instructions on linking an ORCID iD to your Editorial Manager account: https://www.youtube.com/watch?v=_xcclfuvtxQ".

4. Please remove your figures from within your manuscript file, leaving only the individual TIFF/EPS image files, uploaded separately. These will be automatically included in the reviewers’ PDF.

5. Please include your tables as part of your main manuscript and remove the individual files. Please note that supplementary tables (should remain/ be uploaded) as separate ""supporting information"" files".

Reviewers' comments:

Reviewer's Responses to Questions

**Comments to the Author**

1. Is the manuscript technically sound, and do the data support the conclusions?

Reviewer #1: Partly

2. Has the statistical analysis been performed appropriately and rigorously? 

Reviewer #1: Yes

3. Have the authors made all data underlying the findings in their manuscript fully available?

Reviewer #1: No

4. Is the manuscript presented in an intelligible fashion and written in standard English?

Reviewer #1: No

5. Review Comments to the Author

Reviewer #1: This is a systematic review and meta-analysis of clinical studies on nordic walking's effects on walking capacity in PAD and IC.

Comments

1. The way the results are presented across the manuscript, it seems like NW is more beneficial to SET which is not the case. The results are not significant for either MWD or CD.

a. Lines 17-23. Please change the presentation to more simple terms, eg., Compared with SET, NW was NOT associated with an increase in MWD or CD during treadmill or 6-MWT testing in PAD patients. Lines 22-23 are not needed. In other words, the authors should not use wording such as there was an increase in MWD….but not statistically significant. If there is not significance, then there was no increased in MWD between SET and NW….

b. Lines 434-439: same as in lines 17-23… you cannot say that Nordic walking significantly outperformed SET!! Also, no need to report again SMD Z and p values in the discussion. Since you presented this in the results section, it is repetition.

2. Figures 9, 10 and 11, appear opposite to actual results – eg. Figure 9 shows that exercise duration favours NW but results report the opposite. All figures need to be re-checked if correct or not.

3. Line 27: “NW was more effective” might imply that both treatments increase sig. MWD but NW more. However, the data only shows the comparison between the two so we do not know if the control/unsupervised exercise also significantly increased MWD compared to no exercise.

4. Lines 29-31: these refer to a result that is based on one trial, and hence meta-analysis cannot be performed.

5. Same for lines 392-397: they are the same finding reported in the abstract. Since it is based on one trial only, no meta-analysis can be implemented and only qualitatively this result can be discussed. Same for lines 449-452 in discussion.

6. Only one trial used unsupervised NW based on Table 1., which was only used in the MWD between NW and unsupervised exercise therapy. It would be more accurate if the word “supervised” preceded NW (ie. “supervised NW”). NW vs unsupervised exercise findings on MWD are based on 2 trials where one used unsupervised NW and the other supervised NW. This means that the increase in MWD with NW compared to unsupervised exercise therapy may have been influenced by the supervision itself than the exercise modality and this should be discussed in the discussion.

7. Are the results based on week 12 or 24? A bit confused based on line 171.

8. In Table 1, does the N number on sample size include groups not studied in this analyses? Eg. Study by Katarzyna is on n=80 but one group was not used in these analyses.

9. Wherever you refer to ref 27, the name used should be replaced with the author’s last name; ie. instead of “Katarzyna” be replaced with “Kropielnicka”.

10. Please initial the authors that did the data extraction and quality assessment. Eg Lines 143 and 151, next to “two researchers” add the initials of those researchers if they are in the author list.

11. Minor comments:

a. Lines 85-87 needs rephrasing. Since various RCTs have been implemented, it is more accurate to report whether separate RCTs generated conflicting evidence, where a meta-analysis can be helpful in settling such controversies.

b. Please include the year of each of the 4 other meta-analyses presented on page 4. Eg. Cugusi….provided the initial evidence in 2017…etc.

c. Lines 94-97 need to be rephrased.

d. Suggestions: Line 82, “as people with PAD…”; line 72 ”…and uses a specific pair of”; line 97, Secondly is not correct since this is the 3rd meta-analysis?

e. Lines 98-101: this way of reporting actual results is a bit confusing and needs rephrasing.

f. Jansen et al. in a Cochrane meta-analysis and this needs to be mentioned due to their significance in the field of meta-analyses.

g. Lines 106-107: Didn’t 2 of these focus on NW effect on walking in PAD? 17 and 18 references.

h. Rephrase lines 107-109. Also mention up to which years did they cover…It needs to be more clearly stated why a new meta-analysis is needed, since already 4 covered this. Clearly state which years the two meta-analyses that focused on NW covered…

i. Line 171: “exercise were calculated…word duration missing?

j. Line 228: “total sample size…each study’s sample size?

k. Line 289: of each entry, what does that mean?

l. Line 294, limp trials?

m. Line 297, omit “with Bulinska et al”.

6. PLOS authors have the option to publish the peer review history of their article (what does this mean? ). If published, this will include your full peer review and any attached files.

**Do you want your identity to be public for this peer review?** For information about this choice, including consent withdrawal, please see our Privacy Policy .

Reviewer #1: No

---

## [Author Response · Author response to Decision Letter 1]

31 Aug 2024

Thank you very much to the editor and reviewers for giving me the valuable opportunity to revise my manuscript. Your suggestions have greatly benefited me. Regarding all the responses to the view letter, I have submitted them in the rebuttal letter. Thank you.

---

## [Decision Letter · Decision Letter 1]

25 Sep 2024

PONE-D-24-29043R1Effect of Nordic Walking on Walking Ability in Patients with Peripheral Arterial Disease: A Meta-AnalysisPLOS ONE

Dear Dr. Qian,

Thank you for submitting your manuscript to PLOS ONE. After careful consideration, we feel that it has merit but does not fully meet PLOS ONE’s publication criteria as it currently stands. Therefore, we invite you to submit a revised version of the manuscript that addresses the points raised during the review process.

**ACADEMIC EDITOR: ** Please address the points raised by the reviewer below

We look forward to receiving your revised manuscript.

Kind regards,

Shukri AlSaif

Academic Editor

PLOS ONE

Reviewers' comments:

Reviewer's Responses to Questions

**Comments to the Author**

1. If the authors have adequately addressed your comments raised in a previous round of review and you feel that this manuscript is now acceptable for publication, you may indicate that here to bypass the “Comments to the Author” section, enter your conflict of interest statement in the “Confidential to Editor” section, and submit your "Accept" recommendation.

Reviewer #1: (No Response)

2. Is the manuscript technically sound, and do the data support the conclusions?

Reviewer #1: Partly

3. Has the statistical analysis been performed appropriately and rigorously? 

Reviewer #1: Yes

4. Have the authors made all data underlying the findings in their manuscript fully available?

Reviewer #1: (No Response)

5. Is the manuscript presented in an intelligible fashion and written in standard English?

Reviewer #1: Yes

6. Review Comments to the Author

Reviewer #1: Revision comments

1. The authors have now changed the direction of all figures, making results the opposite. I am unsure which is the right direction as they mention that they only changed Figures 9 and 10 but also Figures 5-8 were changed. So now Nordic walking appears better than SET, eg Figure 8. Moreover, Figure 9 now basically shows that claudication distance is lower after SET than NW. I think again the signs/label directions are problematic?

2. Comment 6 response: Perhaps also replacing NW with supervised NW is also suitable in the results section for NW versus SET since no study used unsupervised NW. I understand this is already entered in Table 1, yet it should be also highlighted in the results section and the discussion. I believe unsupervised NW was only used compared to unsupervised therapy (Figure 10) where NW should be left as is. But for the prior results, maybe supervised NW is a better option to describe NW. Eg. lines 373 calling it supervised NW, lines 381, line 389, 398 etc. Same in lines 451, 521, 522, 525. All those places should call NW as supervised NW because otherwise it appears that NW is as effective as SET, yet if both are supervised, this should be mentioned because otherwise it makes NW appear as a better option (if thought as unsupervised).

3. It might be better replacing word “control” in the figures footnotes where Collins et al study is described with SET.

4. Lines 359-362: Please use past tense. Eg. “All included trials obtained approval from the local….” “Participants provided written? Informed consent before enrollment, …”

5. Lines 127-129: Although…introduces a subordinate clause but no main clause follows.

6. While the authors made a great effort to incorporate all reviewer’s suggestions, maybe some of the lines in the introduction could be moved to the discussion. I do appreciate that they added all the new information regarding the 4 prior meta-analyses, perhaps they could be a bit more succinct in the introduction and more expansive in the discussion? The introduction is 5 pages long in the current version. It is highly important to point out what new this meta-analysis adds to the literature beyond the 4 prior ones, yet in a more concise manner.

7. The study by Langbein (REF 24) did not use exercise in control. This has to be entered in Figure 10 and in results section because currently Figure 10 wrongly reports this control as unsupervised ET. Similarly line 418 report it as unsupervised therapy, giving the impression this study did use unsupervised exercise. This is also mentioned in lines 454-455. Same lines 460-463, it appears this study used unsupervised exercise. In fact, pooling these two studies is not that appropriate since the controls were quite different (no exercise vs unsupervised exercise)…If done, it has to be well-defined in the Figure and results and discussion etc. so the reader is fully aware of this. This can be also entered in the limitations section.

8. Lines 463-465: To which two studies do these line refer to?

9. Lines 585-586. This is not really an outcome of this meta-analysis since they are outcomes based on one trial each (one trial using non-exercise and one using unsupervised exercise).

10. Table 1 and Table 2 are larger than the page margins; the authors should make sure it will fit within the journal’s margins.

7. PLOS authors have the option to publish the peer review history of their article (what does this mean? ). If published, this will include your full peer review and any attached files.

**Do you want your identity to be public for this peer review?** For information about this choice, including consent withdrawal, please see our Privacy Policy .

Reviewer #1: No

---

## [Author Response · Author response to Decision Letter 2]

9 Oct 2024

Rebuttal letter

Dear Editor and Reviewer，

Firstly, I would like to sincerely thank the editor and reviewers for taking the time and effort to review my manuscript, which has given me the opportunity to improve its quality. I am very happy about this. The reviewer provided a lot of valuable suggestions in the view letter, and I carefully read and made revisions to each suggestion. During this process, I learned a lot of knowledge and gained a deeper understanding of the exercise method of Nordic walking. All the parts in the manuscript that have been modified according to the suggestions of the reviewer have been marked in red. Now, I will respond to every point raised by the editor and reviewers.

Reviewer’s comments:

1.I am very grateful for every suggestion made by the reviewer. In response to your questions about the figures, I have the following answers. Figure 8 shows the meta-analysis results of supervised NW and SET on claudication distance in PAD patients under 6-MWT testing. This figure shows no statistical difference. The claudication distance refers to the distance that patients with peripheral arterial disease feel pain in their calves or thighs after starting to walk. The longer the claudication distance, the less likely the patient is to feel pain while walking, so a longer claudication distance is better. The effect size of Figure 8 is -0.09, and the standard deviation and mean of the NW group minus the mean and standard deviation of the control group are negative, indicating that claudication distance in the NW group is lower than that in the control group, and claudication distance in the control group is slightly longer. Figure 9 shows the exercise duration of NW compared to SET. The longer the walking time, the better, with an effect size of -0.41, indicating that SET is superior to NW. Therefore, SET has a better exercise effect and prolongs the patient's walking time. This time, I changed the direction of the footnotes in Figure 5-9 back, placing NW on the left and the control group on the right. I consulted some meta-analysis forest plot data and normally placed the intervention group on the left and the control group on the right. The larger the three outcome measures of meta-analysis, the better the patient's walking ability.

2.I fully agree with the reviewer's suggestion and have changed NW to supervised NW as per your request. The revised results can be found in lines 245, 254, 262, 271, 278, 285, 325, 327, 399, 400 and 403 of the manuscript.

3.I greatly appreciate the reviewer's suggestion and have revised the footnote for the control group in the figures to "Control".

4.I am very grateful to the reviewer for pointing out my grammar error. The revised result can be found in lines 227-229 of the manuscript.

5.I am very grateful to the reviewer for pointing out my grammar errors. The revised results are shown in lines 93-96 of the manuscript. The revised content is as follows: Although they provide valuable reference information for selecting appropriate exercise plans for PAD patients in clinical practice by comparing different exercise intervention strategies, these four studies still have limitations in the number of included literature.

6.I am very grateful for the valuable suggestions provided by the reviewer. I have streamlined the introduction according to the suggestions made by the reviewers. I have included the advantages of Nordic walking in the fourth paragraph of the original introduction in the discussion, which is now located in lines 404-414 of the manuscript. Similarly, regarding the four previously published meta-analyses, only the results section of the study was described, located in lines 77-92. The excess content is placed in the first paragraph of the discussion, in lines 295-322 of the manuscript, and a brief explanation of what was added to the previous study is now located in lines 93-106 of the manuscript.

7.I am very grateful for the valuable suggestions provided by the reviewer. After thinking about it, I also believe that two different control groups should not be attributed to unsupervised exercise, as this description is inaccurate. Therefore, I have removed the section comparing the results with unsupervised data and Figure 10, and clearly stated in the Results, Discussion, and Limitations section that meta-analysis cannot be conducted, located at 240-244, 328-342, and 461-464 in the manuscript.

8.I am very grateful to the reviewer for pointing out my negligence. I have now explained in the text which two papers it is, and the results are located in lines 337-340 of the manuscript. The revised content is as follows: It is worth noting that two studies [24] [26] from intervention to week 12 showed that in terms of increasing patient MWD, supervised NW was significantly better than unsupervised NW (1001 ± 913m vs 652 ± 373m).

9.I am very grateful to the reviewer for pointing out my mistake. The conclusion has now been revised, located on lines 469-474 of the manuscript. The revised results are as follows: Nordic walking can improve the walking ability of PAD patients. However, compared to standard exercise therapy (SET), supervised Nordic walking does not have significant advantages; SET was more effective in enhancing exercise duration. Nordic walking presents a viable option when SET is not available. Further high-quality research is needed due to the limited number of studies to validate these findings.

10.Thank you very much for the reviewer's suggestions on the format. The margins of Tables 1 and 2 have been adjusted in the manuscript.

11.Finally, due to some modifications in the results section, I also made corresponding changes to the abstract and sensitivity analysis .

The above is my response to the comments raised by the editor and reviewer. Once again, I would like to express my sincere gratitude. The biggest gain during the submission process was the feedback on the manuscript due to insufficient content. Although there are still many shortcomings in my manuscript, I will work hard to make revisions and strive to meet your requirements. Thank you very much.

Looking forward to letters from editors and reviewers, wishing you a happy life!

Yours sincerely.

Zerong Sun, Yongdong Qian

---

## [Decision Letter · Decision Letter 2]

29 Oct 2024

PONE-D-24-29043R2

Effect of Nordic Walking on Walking Ability in Patients with Peripheral Arterial Disease: A Meta-Analysis

PLOS ONE

Dear Dr. Qian,

Thank you for submitting your manuscript to PLOS ONE. After careful consideration, we feel that it has merit but does not fully meet PLOS ONE’s publication criteria as it currently stands. Therefore, we invite you to submit a revised version of the manuscript that addresses the points raised during the review process.

We look forward to receiving your revised manuscript.

Kind regards,

Shukri AlSaif

Academic Editor

PLOS ONE

Journal Requirements:

Reviewers' comments:

Reviewer's Responses to Questions

**Comments to the Author**

1. If the authors have adequately addressed your comments raised in a previous round of review and you feel that this manuscript is now acceptable for publication, you may indicate that here to bypass the “Comments to the Author” section, enter your conflict of interest statement in the “Confidential to Editor” section, and submit your "Accept" recommendation.

Reviewer #1: (No Response)

Reviewer #2: (No Response)

2. Is the manuscript technically sound, and do the data support the conclusions?

Reviewer #1: Partly

Reviewer #2: Yes

3. Has the statistical analysis been performed appropriately and rigorously? 

Reviewer #1: Yes

Reviewer #2: Yes

4. Have the authors made all data underlying the findings in their manuscript fully available?

Reviewer #1: (No Response)

Reviewer #2: Yes

5. Is the manuscript presented in an intelligible fashion and written in standard English?

Reviewer #1: Yes

Reviewer #2: Yes

6. Review Comments to the Author

Reviewer #1: Thank you for addressing the comments but I still believe there is an issue with the forest plots directions.

For example, in Figure 5, the first study by Bulinska has a mean of 364 in Nordic walking versus 290 in SET. This means that the difference is favouring Nordic walking by 74m in max walking distance. However, the plot shows that it favours the control. For Collins study, it is 612 m in Nordic vs 889 in SET, hence SET patients walked on average more than Nordic by 277m, but the plot shows that a favoring for Nordic walking while it should be showing SET. Same for the rest of studies.

Then the same applies to Figure 6, Figure 7, Figure 8 and Figure 9.

As an example, please check how forest plots directions are in the meta-analysis by Parmenter et al. 2020, “Resistance training as a treatment for older persons with peripheral artery disease: a systematic review and meta-analysis” Br J Sports Med 2020;54:452–461. doi:10.1136/bjsports-2018-100205

Here to also note that label Control in the plot and footnotes would be more informative if it was replaced by the word “SET”.

Reviewer #2: The authors undertook to investigate how Nordic walking affects the ability to walk in patients with PAD. Publications covering treadmill tests and 6-MWT were included in the meta-analysis. The selection of manuscripts was carried out correctly. The duration of the intervention, ranging from 12 to 24 weeks, was also taken into account. Indicating that NW improves the patient's MWD compared to other treatment methods (except SET), but it depends on the monitoring itself, not on the exercise style.

The discussion was prepared correctly and includes the available literature.

In conclusion, I propose to accept the publication in its current form.

7. PLOS authors have the option to publish the peer review history of their article (what does this mean? ). If published, this will include your full peer review and any attached files.

**Do you want your identity to be public for this peer review?** For information about this choice, including consent withdrawal, please see our Privacy Policy .

Reviewer #1: No

Reviewer #2: **Yes: ** Artur Kruszewski

---

## [Author Response · Author response to Decision Letter 3]

5 Nov 2024

Rebuttal letter

Dear Editor and Reviewer，

Firstly, I would like to sincerely thank the editor and reviewers for taking the time and effort to review my manuscript, which has given me the opportunity to improve its quality. I am very happy about this. The reviewer provided a lot of valuable suggestions in the view letter, and I carefully read and made revisions to each suggestion. During this process, I learned a lot of knowledge and gained a deeper understanding of the exercise method of Nordic walking. All the parts in the manuscript that have been modified according to the suggestions of the reviewer have been marked in red. Now, I will respond to every point raised by the editor and reviewers.

Editor's comment：

1.Thank you for the editor's reminder. I have checked the format of the references and highlighted the changes in red. I have added the month of publication, for example, 2024; 45 (15): 1303-1321 is modified to 2024 Apr14; 45(15):1303-1321.

Reviewer’s comments:

1.Thank the reviewer for raising my question. I also realized that there was an error in the footnotes of my forest plot. After reading the meta-analysis you provided, I made changes in the figures, and I also consulted relevant materials. When creating a forest plot with RevMan software, the default research event of the software is "adverse event", such as incidence rate, mortality or negative results. However, if it is a “favorable event”, the default setting of the software needs to be modified. For example, the comprehensive effect size of Figure 5 is -0.06, and the forest plot is slightly biased towards the left of the invalid line, while the result is a favorable event. Therefore, the left footnote should be modified to SET and the right to NW, and the same applies to other graphs. Thank you again to the reviewer for their valuable feedback.

2.Thank you very much for the second reviewer's affirmation of my article.

3.Finally, I only made modifications in the figures and references, and did not make any other changes. Thank you to the editor and reviewers.

The above is my response to the comments raised by the editor and reviewer. Once again, I would like to express my sincere gratitude. The biggest gain during the submission process was the feedback on the manuscript due to insufficient content. Although there are still many shortcomings in my manuscript, I will work hard to make revisions and strive to meet your requirements. Thank you very much.

Looking forward to letters from editors and reviewers, wishing you a happy life!

Yours sincerely.

Zerong Sun, Yongdong Qian

---

## [Editor Report · Decision Letter 3]

14 Nov 2024

PONE-D-24-29043R3Effect of Nordic Walking on Walking Ability in Patients with Peripheral Arterial Disease: A Meta-AnalysisPLOS ONE

Dear Dr. Qian,

Thank you for submitting your manuscript to PLOS ONE. After careful consideration, we feel that it has merit but does not fully meet PLOS ONE’s publication criteria as it currently stands. Therefore, we invite you to submit a revised version of the manuscript that addresses the points raised during the review process.

PLEASE ADDRESS THE ISSUE WITH THE FOREST PLOT.

We look forward to receiving your revised manuscript.

Kind regards,

Shukri AlSaif

Academic Editor

PLOS ONE
---

## [Author Response · Author response to Decision Letter 4]

19 Nov 2024

Rebuttal letter

Dear Editor and Reviewer，

Firstly, I would like to sincerely thank the editor and reviewers for taking the time and effort to review my manuscript, which has given me the opportunity to improve its quality. I am very happy about this. The reviewer provided a lot of valuable suggestions in the view letter, and I carefully read and made revisions to each suggestion. During this process, I learned a lot of knowledge and gained a deeper understanding of the exercise method of Nordic walking. All the parts in the manuscript that have been modified according to the suggestions of the reviewer have been marked in red. Now, I will respond to every point raised by the editor and reviewers.

Editor's and Reviewer's comment：

1.Thank you very much to the editor for giving me another opportunity to revise the manuscript. I am very grateful.

Regarding the issue of resolving the forest plot, I carefully checked the footnotes of my forest plot and it should be fine. In the previous revision, I followed the reviewer's suggestion and changed the left side of the forest plot footnote to the control group, which is the SET group, and the right side to the intervention group, which is the NW group. The outcome measure of this study aims to encourage patients to increase their walking distance as much as possible, in order to achieve favorable outcomes. Therefore, the default direction automatically generated by Revman software needs to be modified, and I also referred to forest plots from other journals, as shown in the following figure. If you have any other suggestions, I am happy to make revisions.（The forest plot of resistance training and treadmill walking training is shown below, with an effect size of -0.47, indicating that the results support the treadmill group）。

2.I have revised the format of the references based on the editor's suggestion and highlighted them in red. Additionally, I have replaced reference 16 as the revised article cannot read the entire text.

3.I have uploaded the pictures to PACE and all the pictures meet the requirements.

4.I have added supporting information after the conclusion of the manuscript, and all data in this study are included in the supporting information folder.

The above is my response to the comments raised by the editor and reviewer. Once again, I would like to express my sincere gratitude. The biggest gain during the submission process was the feedback on the manuscript due to insufficient content. Thank you very much.

Looking forward to letters from editors and reviewers, wishing you a happy life!

Yours sincerely.

Zerong Sun, Yongdong Qian

---

## [Decision Letter · Decision Letter 4]

6 Dec 2024

Effect of Nordic Walking on Walking Ability in Patients with Peripheral Arterial Disease: A Meta-Analysis

PONE-D-24-29043R4

Dear Dr. Qian,

We’re pleased to inform you that your manuscript has been judged scientifically suitable for publication and will be formally accepted for publication once it meets all outstanding technical requirements.

Kind regards,

Shukri AlSaif

Academic Editor

PLOS ONE

Additional Editor Comments (optional):

Reviewers' comments:

Reviewer's Responses to Questions

**Comments to the Author**

1. If the authors have adequately addressed your comments raised in a previous round of review and you feel that this manuscript is now acceptable for publication, you may indicate that here to bypass the “Comments to the Author” section, enter your conflict of interest statement in the “Confidential to Editor” section, and submit your "Accept" recommendation.

Reviewer #1: All comments have been addressed

2. Is the manuscript technically sound, and do the data support the conclusions?

Reviewer #1: Yes

3. Has the statistical analysis been performed appropriately and rigorously? 

Reviewer #1: Yes

4. Have the authors made all data underlying the findings in their manuscript fully available?

Reviewer #1: (No Response)

5. Is the manuscript presented in an intelligible fashion and written in standard English?

Reviewer #1: Yes

6. Review Comments to the Author

Reviewer #1: (No Response)

7. PLOS authors have the option to publish the peer review history of their article (what does this mean? ). If published, this will include your full peer review and any attached files.

**Do you want your identity to be public for this peer review?** For information about this choice, including consent withdrawal, please see our Privacy Policy .

Reviewer #1: No

---

## [Editor Report · Acceptance letter]

PONE-D-24-29043R4

PLOS ONE

Dear Dr. Qian,

I'm pleased to inform you that your manuscript has been deemed suitable for publication in PLOS ONE. Congratulations! Your manuscript is now being handed over to our production team.

Kind regards,

on behalf of

Dr. Shukri AlSaif

Academic Editor

PLOS ONE